# Comprehensive Analysis of CDK1-Associated ceRNA Network Revealing the Key Pathways LINC00460/LINC00525-Hsa-Mir-338-FAM111/ZWINT as Prognostic Biomarkers in Lung Adenocarcinoma Combined with Experiments

**DOI:** 10.3390/cells11071220

**Published:** 2022-04-04

**Authors:** Wen Li, Shan-Shan Feng, Hao Wu, Jing Deng, Wang-Yan Zhou, Ming-Xi Jia, Yi Shi, Liang Ma, Xiao-Xi Zeng, Zavuga Zuberi, Da Fu, Xiang Liu, Zhu Chen

**Affiliations:** 1College of Life Sciences and Chemistry, Hunan University of Technology, Zhuzhou 412007, China; T20202532@csuft.edu.cn (W.L.); m19077700015@stu.hut.edu.cn (S.-S.F.); T20202509@csuft.edu.cn (J.D.); claralin527@hut.edu.cn (L.M.); zengxiaoxi@hut.edu.cn (X.-X.Z.); 2National Engineering Research Center of Rice and Byproduct Deep Processing, College of Food Science and Engineering, Central South University of Forestry and Technology, Changsha 410004, China; 20213551@csuft.edu.cn (H.W.); 20200100079@csuft.edu.cn (M.-X.J.); 20210100081@csuft.edu.cn (Y.S.); 3Department of Medical Record, Hengyang Medical School, The First Affiliated Hospital, University of South China, Hengyang 421001, China; 2018010013@mail.usc.edu.cn; 4Department of Science and Laboratory Technology, Dar es Salaam Institute of Technology, Dar es Salaam P.O. Box 2958, Tanzania; zavuga.zuberi@dit.ac.tz; 5Central Laboratory for Medical Research, Shanghai Tenth People’s Hospital, Tongji University School of Medicine, Shanghai 200072, China; fuda@tongji.edu.cn; 6Department of Thoracic Surgery, Hengyang Medical School, The Second Affiliated Hospital, University of South China, Hengyang 421001, China

**Keywords:** Lung adenocarcinoma, ceRNA network, CDK1 gene, LINC00525/LINC00460, FAM111B/ZWINT gene, prognosis

## Abstract

Lung adenocarcinoma (LUAD) is the leading cause of cancer deaths worldwide, and effective biomarkers are still lacking for early detection and prognosis prediction. Here, based on gene expression profiles of LUAD patients from The Cancer Genome Atlas (TCGA), 806 long non-coding RNAs (lncRNAs), 122 microRNAs (miRNAs) and 1269 mRNAs associated with CDK1 were identified. The regulatory axis of LINC00460/LINC00525-hsa-mir-338-FAM111B/ZWINT was determined according to the correlation between gene expression and patient prognosis. The abnormal up-regulation of FAM111B/ZWINT in LUAD was related to hypomethylation. Furthermore, immune infiltration analysis suggested FAM111B/ZWINT could affect the development and prognosis of cancer by regulating the LUAD immune microenvironment. EMT feature analysis suggested that FAM111B/ZWINT promoted tumor spread through the EMT process. Functional analysis showed FAM111B/ZWINT was involved in cell cycle events such as DNA replication and chromosome separation. We analyzed the HERB and GSCALite databases to identify potential target medicines that may play a role in the treatment of LUAD. Finally, the expression of LINC00460/LINC00525-hsa-mir-338-FAM111B/ZWINT axis was verified in LUAD cells by RT-qPCR, and these results were consistent with bioinformatics analysis. Overall, we constructed a CDK1-related ceRNA network and revealed the LINC00460/LINC00525-hsa-mir-338-FAM111/ZWINT pathways as potential diagnostic biomarkers or therapeutic targets of LUAD.

## 1. Introduction

Lung cancer is the second most common cancer and the main cause of cancer death in the world [1]. The incidence of lung cancer has been increasing because of factors such as smoking, air pollution and occupational exposure. In addition, due to the lack of obvious clinical manifestations and early diagnosis, the 5-year overall survival (OS) rate is about 17%. At present, the main treatment methods for lung cancer include surgery, radiotherapy, chemotherapy, immunotherapy, targeted therapy, etc. Surgical resection of cancer tissue is the first choice of treatment with early-stage lung cancer. Radiotherapy is also an important treatment; stereotactic radiotherapy locates the tumor through three-dimensional stereo imaging technology, and then applies radiation therapy to reduce the damage to normal parts [2]. With the development of cancer, cancer cells are widely transferred to lymph nodes and other organs and tissues of the whole body, and it is difficult to completely eliminate cancer cells by local resection and radiotherapy. Chemotherapy and immunotherapy are effective means for systemic treatment and standard treatment of lung cancer. Common anticancer drugs include Paclitaxel, Docetaxel and Vinorelbine, and CTLA-4 and PD-1 checkpoint inhibitors [3,4]. While drugs kill cancer cells, they also cause some damage to normal cells. Moreover, the resistance of cancer cells often leads to the failure of drug treatment [4,5,6]. Molecular targeted therapy, also known as “biological missile”, is more specific than traditional broad-spectrum anticancer therapy, and compared with chemotherapy, targeted drugs have less side effects on normal cells [7]. Even though more and more molecular targeted therapies have shown good progress, the current clinical treatment of Lung adenocarcinoma (LUAD) is not optimistic. Therefore, understanding the pathogenesis and identifying new diagnostic, prognostic and drug resistance biomarkers are important for the study of LUAD.

The functions of long non-coding RNAs (lncRNAs) have been proved in protecting chromosome integrity, maintaining genome structure, transcription, translation and epigenetic regulation [8,9,10]. Imbalances of lncRNA are also associated with the development of many diseases and complications, such as premature preterm rupture of membrane, autism spectrum disorders, and cancer [11,12,13,14]. MicroRNAs (miRNAs) target and induce translation inhibition or degradation of mRNA to control gene expression and regulate biological processes [15,16]. They play an important role in tumorigenesis and metastasis by targeting multiple oncogenic signaling pathways such as Notch, Wnt/β-catenin, JAKs/STAT3, PI3K/AKT and NF-κB pathway [17,18]. Professor Salmena believed mRNAs and lncRNAs “talk” to each other using miRNA response elements (MREs), and form a large-scale regulatory network—the ceRNA network (Appendix A) [19]. ceRNAs are involved in the pathogenesis of many cancers, such as ovarian, gastric and liver cancer [20,21,22,23,24].

The proliferation of cancer cells is related to the imbalance of cyclin expression [25,26]. CDK1 is an important cell cycle regulatory protein, which is a key factor in the G2/M phase transition in cell mitosis [27,28,29]. High expression of CDK1 is related to the occurrence and poor prognosis of colorectal, liver, lung and pancreatic cancer [30,31,32,33,34]. However, CDK1 inhibitors can prevent cell division and promote cancer cell apoptosis and tumor regression [35,36]. The knockdown of CDK1 mediated by ShRNA increases cellular sensitivity to chemotherapeutic agents and the apoptosis of ovarian cancer cell lines [37].

In this study, gene expression profiles and clinical information of LUAD patients in The Cancer Genome Atlas (TCGA) database were obtained and comprehensively analyzed, and a CDK1-related ceRNA network was constructed. The triple regulatory network of LINC00460/LINC00525-hsa-mir-338-FAM111/ZWINT was identified by expression analysis and survival analysis. Additionally, the multi-gene interaction regulation analysis model was used to analyze the correlation between different expression patterns and OS of LUAD patients. Finally, the expression of LINC00460, LINC00525, hsa-mir-338, FAM111 and ZWINT LUAD cell lines was verified by RT-PCR. Therefore, our research is helpful to understand the molecular mechanism of LUAD and provide a new target for LUAD treatment.

## 2. Materials and Methods

### 2.1. Expression of CDK1 in LUAD

Expression and pairwise analysis of CDK1 in pan-cancer samples and normal samples were performed using the TCGA database. Gene expression profile interaction analysis 2 (GEPIA2, http://gepia2.cancer-pku.cn/, accessed on 23 June 2021) and Human Protein Atlas (HPA, http://www.proteinatlas.org/, accessed on 23 June 2021) were used to analyze CDK1 transcription and protein expression in LUAD cell lines.

### 2.2. Preparation of LUAD Data

RNA sequencing data were obtained from the TCGA database (https://portal.gdc.cancer.gov/, accessed on 29 June 2021). The sample exclusion criteria were (i) histological diagnosis negating LUAD, (ii) presence of malignancy other than LUAD, (iii) lack of complete clinical data, and (iv) inability to match RNA and miRNA samples in the database. A total of 515 LUAD tissue samples were used in this study. Patient clinical data also originated from TCGA database and did not require ethics committee approval. This study complied with the publication guidelines provided by TCGA.

### 2.3. Gene Expression Analysis

According to the above description, the high expression of CDK1 was associated with the development of LUAD, so we constructed a CDK1-related ceRNA network as a potential prognostic model for LUAD. According to CDK1 level, we divided the LUAD samples into low CDK1 expression group (*n* = 257) and high CDK1 expression group (*n* = 258). log_2_FC and *p* values of each gene were calculated by R package “gplots”. Threshold values were set for lncRNA (|log_2_FC| > 1.2), miRNA (|log_2_FC| > 0.8) and mRNA (|log_2_FC| > 1.2), and the level of the RNAs was considered statistically different when the *p* value < 0.05. GraphPad Prism version 7.04 was used to display the volcano plots of differentially expressed RNAs (DERNAs).

Mircode (http://www.mircode.org/, accessed on 30 June 2021) was used to predict the mutual regulation between DElncRNAs and DEmiRNAs, and then targetscan (http://www.targetscan.org/, accessed on 30 June 2021) and miRDB (http://www.mirdb.org/mirdb/, accessed on 30 June 2021) databases were used to predict the target genes of DEmiRNAs and obtain the intersection with DEmRNAs. CDK1-related ceRNA network was visualized by Cystocape version 3.7.1. In order to understand the possible biological processes and pathways of the network, we performed a gene ontology (GO) enrichment analysis of the DEmRNAs in the ceRNA network in Metascape (http://metascape.org/gp/index.html, accessed on 2 July 2021).

### 2.4. Survival Analysis of ceRNA Network

Clinical information of patients was downloaded from the TCGA database. Then, univariate Kaplan-Meier analysis was performed on the matched data to determine the relationship between genes and OS. The survival-related ceRNA network was a visual network based on survival-related DElncRNAs, DEmiRNAs and DEmRNAs.

### 2.5. Construction of a Prognostic Model for LUAD

The CDK1-related ceRNA network was screened by analyzing the expression level of DERNAs in the same patient LUAD and normal samples as well as CDK1^low^ and CDK1^high^ groups. In addition, pairwise analysis of lncRNA-miRNA, miRNA-mRNA, lncRNA-mRNA and DEmRNAs-CDK1 was performed to expose the correlation of expression between DERNAs. The cellular localization of lncRNAs determines their potential mechanisms; the subcellular localization of lncRNAs was analyzed by LNCipedia (https://lncipedia.org/, accessed on 5 July 2021). Base pairing among LINC00460/LINC00525-has-mir-338-FAM111B/ZWINT was predicted by MiRcode, lncRNABase and TargetScan.

### 2.6. Methylation and Phosphorylation Analysis and Mutation Analysis

DNA methylation was one of the first modification pathways identified. To investigate the mechanism of aberrant up-regulation of FAM111B and ZWINT in LUAD, the methylation levels of target genes were assessed using the UALCAN database (http://ualcan.path.uab.edu/, accessed on 7 July 2021). The Beta value indicated the DNA methylation level in the range from 0 (unmethylated) to 1 (fully methylated). We also used the MEXPRESS (http://mexpress.be, accessed on 7 July 2021) and Methsurv (https://biit.cs.ut.ee/Methsurv/, accessed on 7 July 2021) databases to query the methylation sites and associated differentially methylated regions. The CPTAC dataset from the UALCAN database (http://ualcan.path.uab.edu/, accessed on 7 July 2021) was used to analyze the phosphorylation level of proteins expressed by the target genes. Z-values represent the standard deviation of the median between samples for LUAD, and Log_2_ spectral count rate values for samples were normalized within profiles and between samples.

The cBioPortal dataset (https://www.cbioportal.org/, accessed on 9 July 2021) contained multiple data types such as somatic mutations, DNA methylation, protein enrichment, and miRNA expression to facilitate the study of multidimensional cancer gene datasets. The mutation of FAM111B and ZWINT in tumor was visualized by cBioPortal.

### 2.7. Immune Infiltration Levels

TIMER (https://cistrome.shinyapps.io/timer/, accessed on 9 July 2021) is an online tool focused on the quantification of immune infiltration of cells. The “SCNA” module provided a comparative study of tumor infiltration levels among tumors with various somatic copy number changes of a given gene. The “Survival” module showed the clinical relevance of immune cells in the tumor. In addition, the “GENE” module visualized the correlation between the genes of interest and the level of immune infiltration in different cancers. Finally, we used the “Correlation” module to predict the correlation of FAM111B and ZWINT genes with 16 markers of immune cells.

### 2.8. Functional Enrichment Analysis and Interaction Networks

To investigate the possible biological processes and pathways involved in the screened LUAD prognostic models, the top 100 genes associated with CDK1, FAM111B and ZWINT were obtained from GEPIA (http://gepia.cancer-pku.cn/, accessed on 9 July 2021). We used the R package “clusterProfiler” to perform GO and Kyoto encyclopedia of genes and genomes (KEGG) pathway enrichment analysis on these genes and selected the top 5 most relevant pathways. The enrichment was verified by DAVID (https://david.ncifcrf.gov/tools.jsp, accessed on 7 July 2021) and visualized by “ggplot2”, an ontology-based R package.

GeneMINIA (http://genemania.org/, accessed on 9 July 2021) contains co-expression, genetic interaction, protein-protein physical interaction, shared protein domains, co-localization and pathway data. A unique scoring function was used to predict the functional association network of target genes. STRING database (https://string-db.org/, accessed on 10 July 2021) maps the interactions network based on protein co-promotion functions.

### 2.9. Targeted Medicines Analysis

HERB database (http://herb.ac.cn/, accessed on 17 July 2021) is a natural herbal medicine database platform, which was used to query possible targeted herbs. GSCAlite (http://bioinfo.life.hust.edu.cn/web/GSCALite/, accessed on 2 August 2021) is a gene set cancer analysis platform. In this platform, Spearman was used to analyze the correlation between 481 drug probes in CTRB and genes. In addition, the same analysis was conducted between genes and more than 100 compounds in the Genomics of Drug Sensitivity in Cancer (GDSC) database.

### 2.10. Extraction of RNA from Cells In Vitro and Verification of RT-qPCR

One normal lung epithelial cell line (BEAS-2B) and three LUAD cell lines (A549, PC-9 and H1299) were provided for RNA extraction and quantitative PT-PCR for gene expression analysis from Xiangya Hospital of Central South University (Changsha, Hunan Province, China). Cells were cultured in RPMI1640 medium (Gibco, New York, NY, USA) with 10% fetal bovine serum (Gibco, New York, NY, USA) and 1% penicillin-streptomycin (Gibco, New York, NY, USA) medium in a constant temperature and humidity incubator with 5% carbon dioxide at 37 °C. Cells were digested with trypsin (Gibco, New York, NY, USA). Total RNA was extracted by FastPure Cell/Tissue Total RNA Isolation Kit (Vazyme, Nanjing, Jiangsu Province, China). The reverse transcription process of mRNA and lncRNA was carried out according to the instructions of BeyoRT™ II cDNA Synthesis Kit (Beyotime, Shanghai, China). The miRNA 1st Strand cDNA Synthesis Kit (Vazyme, Nanjing, Jiangsu Province, China) was used for reverse transcription of miRNA by the stem-loop method. Lastly, SYBR Green PCR Kit (Beyotime, Shanghai, China) was used for quantitative polymerase chain reaction (q-PCR) reaction. GAPDH and U6 were used as internal reference genes. The relative expression was calculated by ΔΔ Ct method. The primer sequences are shown in Appendix A.

### 2.11. Statistical Analysis

Data were visualized using GraphPad Prism version 7.04, Cystocape version 3.7.1, and analyzed by SPSS 23.0 software (SPSS Inc, Chicago, IL, USA). Differences between the two datasets were calculated using the Mann-Whitney U test and Wilcoxon signed rank test, and univariate Kaplan-Meier analysis was used to detect the relationship between gene expression and patient OS. *p* < 0.05 was considered significantly different.

## 3. Results

### 3.1. The High Expression of CDK1 in LUAD and Its Correlation with the Prognosis of Patients

Paired analysis showed that the expression level of CDK1 in most cancer tissues was higher than that in normal tissues in LUAD (Figure 1A–C). A summary of specific values is disclosed in Appendix A. The HPA database verified that CDK1 was significantly up-regulated in LUAD at the protein level (Figure 1D). Based on the relationship between CDK1 expression and OS or TNM staging in GEPIA2, these results showed that high expression of CDK1 reduced OS in LUAD (Figure 1E) and led to a worse prognosis (Figure 1F). Moreover, the clinical information of patients was analyzed. The results showed that the expression of CDK1 in male patients was significantly higher than in female patients (Appendix A), significantly higher in patients younger than those older than 65 years (Appendix A), and also significantly higher in smokers than in nonsmokers (Appendix A). The expression of CDK1 was related to sex, age and smoking, but not to race (Appendix A). In summary, these data indicated that CDK1 expression is up-regulated in LUAD and related to the development of cancer.

### 3.2. CDK1-Related DERNAs in LUAD and Construction of ceRNA Network

To construct a CDK1-related ceRNA network, the samples were divided into CDK1^low^ group (*n* = 257) and CDK1^high^ group (*n* = 258) according to the expression of CDK1, and the log_2_Fold Change (log_2_FC) and *p* values of the differential genes were calculated by the R package “gplots”. By setting the lncRNA threshold |log_2_FC| > 1.2, miRNA threshold |log_2_FC| > 0.8, and mRNA threshold |log_2_FC| > 1.2, *p* value < 0.05 indicated that the data were significantly different. A total of 806 lncRNAs, 122 miRNAs and 1269 mRNAs were screened (Figure 2A–C). Volcano maps were generated with GraphPad Prism version 7.04.

In order to obtain the regulatory relationship between DERNAs, based on miRcode, TargetScan and miRDB databases, 53 DElncRNAs, 7 DEmiRNAs and 47 DEmRNAs were used to construct and visualize the ceRNA network (Figure 2D). GO enrichment analysis using Metascape showed that DEmRNAs were enriched in “histone deacetylase binding”, “synaptic membrane” and “sodium ion transmembrane transporter activity” (Figure 2E).

### 3.3. Construction of CDK1 Expression-Related Prognostic Model for LUAD

According to univariate Kaplan-Meier regression analysis and inter-matching of lncRNA-miRNA-mRNA, 6 DElncRNAs, 2 DEmRNAs, and 5 survival-related DEmRNAs were used to construct the survival-related ceRNA network (Appendix A).

To construct a prognostic model for CDK1 expression-related LUAD, we further analyzed the expression levels of DERNAs in CDK1^low^ and CDK1^high^ groups, and *p* < 0.05 was considered statistically significant (Appendix A). The results demonstrated that TDRG1, LINC00221, and AP002478.1 had no reference value due to low expression. Further, the expression of MACROD2 did not conform to the regulatory network of lncRNA-miRNA-mRNA. The remaining 7 DERNAs were paired for expression patterns and analyzed for expression levels in tumor tissues and normal tissues of LUAD patients (Appendix A). The result showed that H19 was undifferentiated (*p* > 0.05). Through paired analysis, there was no correlation between IL36RN and hsa-mir-338. Interestingly, a similar result was gained between IL36RN and LINC00460 (Appendix A). Besides, hsa-mir-338 was negatively correlated with the expression of LINC00460 and LINC00525, while two lncRNAs (LINC00460 and LINC00525) showed positive correlation with the mRNAs (FAM111B and ZWINT). The above results suggested that LINC00460 and LINC00525 could improve the levels of FAM111B and ZWINT via hsa-mir-338 sponge. The expression pairing analysis between DERNAs and CDK1 revealed a negative correlation between hsa-mir-338 and CDK1. Contrarily, the correlations between LINC00460, LINC00525, FAM111B, and ZWINT and CDK1 were positive (Appendix A).

Investigation of the cell localization of lncRNAs based on LNCipedia showed that the LINC00460 and LINC00525 were predominantly distributed in the cytoplasm (Figure 3A,B). The results showed that LINC00460 and LINC00525 might relieve repression of FAM111B and ZWINT via hsa-mir-338 sponge. Therefore, we determined the ceRNA regulatory network of LINC00460/LINC00525 (up-regulated)-hsa-mir-338 (down-regulated)-FAM111B/ZWINT (up-regulated) (Figure 3C). In addition, MiRcode, lncRNABase and TargetScan were used to predict the base pairing between has-mir-338 and LINC00460/LINC00525 and FAM111B/ZWINT (Figure 3D).

### 3.4. Expression of FAM111B and ZWINT in Pan-Cancer

To understand the expression of the prognostic model in pan-cancer, the TCGA database was utilized to analyze the expression of DERNAs (some cancers were not considered due to the small sample size). As shown in Appendix A, LINC00460, LINC00525, FAM111B and ZWINT were upregulated and hsa-miR-338 was downregulated in most cancers. Furthermore, there was a similar expression trend in breast invasive carcinoma, lung squamous cell carcinoma, thyroid carcinoma, etc. They also composed the ceRNA regulatory network, which was LINC00460/LINC00525 (up-regulated)-hsa-mir-338 (down-regulated)-FAM111B/ZWINT (up-regulated).

### 3.5. Methylation and Phosphorylation Analysis of FAM111B and ZWINT

Gene expression is a complex process regulated by multiple factors [38]. To explore the mechanism of abnormal upregulation of FAM111B and ZWINT in LUAD, methylation level analysis was performed using the UALCAN database. The results showed that FAM111B and ZWINT were significantly lower in LUAD methylation levels than normal cells (Appendix A). DNA methylation is an important DNA modification that regulates the proper silencing of genes [39]. We obtained the methylation site cg2097987 negatively correlated with FAM111B expression, which was located in N_Shore, TSS1500 region (Appendix A). Similarly, two methylation sites (cg14859667 and cg14842833) negatively correlated with ZWINT were located in CpG Island, 5′ UTR and S_shore, TSS 1500, respectively (Appendix A). The above findings suggested that methylation may be involved in the regulation of FAM111B/ZWINT expression, and then affect the development of LUAD.

Actually, abnormal protein phosphorylation has been linked to many diseases [40]. To investigate the role for phosphorylation in regulating protein activity, based on the UALCAN database, we found that FAM111B and ZWINT phosphorylation levels were up-regulated in LUAD (Appendix A).

### 3.6. Mutations of FAM111B and ZWINT

Mutation plays an important role in the occurrence of cancer [41]. According to an analysis of the TCGA database using the cBioPortal tool, the mutation rates of FAM111B and ZWINT in LUAD were 1.8% and 3%, and missense was the main mutation type (Appendix A). In addition, the “Mutations” module was used to query common mutation sites. The results indicated that FAM111B had six common missense sites. Similarly, ZWINT had ten common missense sites, three truncating sites and two splice sites (Appendix A). In order to obtain the correlation between mutation and OS, we used the “Comparison/Survival” module, and the results showed that FAM111B and ZWINT were not correlated with patient OS (Appendix A). Using cbioportal tool, the mutations of FAM111B and ZWINT in 3513 patient samples were compared with those closely related to lung cancer driver genes from 10 databases, including TCGA (Appendix A) [42,43,44,45,46,47]. The results demonstrated that there was no direct mutual exclusion or co-expression relationship between mutations of FAM111B and ZWINT and common mutations of lung cancer. Interestingly, EGFR mutation was mutually exclusive with mutations of KRAS, LRP1B, ALB, BRAF and other genes, but it did co-occur with the mutation of TP53. In addition, the mutation of KRAS was mutually exclusive with the mutations of genes such as ERBB2, TP53 and BRAF.

Finally, pan-cancer mutation analysis showed that FAM111B and ZWINT had the highest mutation rates in u corpus endometrial carcinoma and stomach adenocarcinoma, respectively (Appendix A).

### 3.7. Levels of Immune Infiltration of FAM111B and ZWINT

To investigate the immune infiltration levels of FAM111B and ZWINT in cancer, firstly, the “SCNA” module was used to evaluate the relationship between the copy number of genes and the infiltration level of immune cells. The figure showed that the gene copy number of FAM111B correlated with the infiltration levels of B cells, CD4^+^ T cells, macrophages, neutrophils, and dendritic cells (Appendix A). Meanwhile, ZWINT was associated with the levels of B cell, CD4^+^ T cell, and dendritic cell infiltration (Appendix A). Moreover, further assessing the impact of immune cells on patient OS, the “Survival” module showed that high levels of B cells and dendritic cells in LUAD resulted in a better prognosis (Appendix A). Finally, the “Gene” module indicated that FAM111B was positively correlated with neutrophils and dendritic cells. However, the opposite result was obtained between FAM111B and CD4^+^ T cells (Appendix A). ZWINT was negatively correlated with B cells and macrophages (Appendix A). In addition, we used the R package “ggplot2” to correlate the expression of FAM111B and ZWINT with immune checkpoints and immune activation (Appendix AA–D) [48,49]. The results demonstrated that the high levels of FAM111B and ZWINT were related to the high expression of several immune checkpoints. Tumor mutational burden (TMB) could be used as a biomarker of immunotherapy, and patients with high TMB are more likely to benefit from immunotherapy [50]. In order to evaluate the correlation between gene expression level and TMB, we downloaded the mutation data of LUAD from TCGA, and counted the TMB value of patient samples. The results showed that the high expression of FAM111B and ZWINT was related to the high TMB (Appendix A). Finally, the “Correlation” module was applied to search for the relationship between FAM111B/ZWINT and 16 immune cell markers (Appendix A). The results indicated that the expression of FAM111B was positively correlated with immune markers of CD8^+^ T cells, monocytes, TAM and other cells. At the same time, it was also relevant to some immune markers of cells such as M1 macrophages and M2 macrophages. Moreover, ZWINT correlated positively with immune markers of CD8^+^ T cells and B cells and negatively with neutrophils. We also used the GEPIA database to check the relationship between FAM111B and ZWINT and immune markers (Appendix A). The results were similar to those of the TIMER database query.

Because of the interaction between tumor and stromal components, many immune cells will chemotaxis to this point, forming a complex tumor immune microenvironment (TIME) together. The heterogeneity of TIME makes the progress of tumors vary greatly among individuals. The TIME score between samples was calculated by ESTIMATE software and compared with the expression of FAM111B/ZWINT and the OS of patients (Appendix A). The results showed that the low immune score was related to the poor prognosis of patients. Furthermore, high expression of ZWINT was associated with low immune score. In a word, it was suggested that FAM111B/ZWINT may promote the development of LUAD and prognosis by regulating the levels of immune cell infiltration.

### 3.8. Epithelial-Mesenchymal Transition (EMT) Analysis

EMT, as the culprit of tumor spread, changed the cell morphology and molecular marker level during this process, and gained stronger invasion and metastasis ability [51]. The loss of epithelial marker and the development of interstitial marker are considered indications that the cell has experienced EMT. In this paper, we compared the levels of FAM111B and ZWINT with EMT markers (Appendix A). The results showed that the expression of FAM111B and ZWINT was negatively correlated with epithelial marker Mucin-1 (MUC1) and positively correlated with mesenchymal markers N-cadherin (CDH2) and Fibronectin (FN1). It intimated that FAM111B and ZWINT may promote tumor spread through the EMT process.

### 3.9. Gene Enrichment Analysis and Interaction Networks

To understand the biological processes and pathways that may be involved in the prognostic model of LUAD, the first 100 genes related to CDK1, FAM111B and ZWINT were enriched and analyzed with R package “clusterProfiler” (Figure 4A–C). In addition, a Venn diagram was drawn by R package “ggplot2” (Figure 4D). KEGG pathway analysis revealed that CDK1, FAM111B and ZWINT were enriched in cell cycle, DNA replication. Interestingly, all of the pathways were associated with cell proliferation. Meanwhile, GO enrichment analysis showed that the biological processes (BP) of CDK1, FAM111B and ZWINT were enriched in chromosome segregation and nuclear division, and the cellular components (CC) were enriched in chromosomal region, centromeric region and spindle. In addition, the molecular functions (MF) were enriched in microtubule binding. Enrichment analysis with DAVID verified that the pathways of CDK1, FAM111B and ZWINT were mostly duplicated and associated with cell cycle (Appendix A). The above findings suggested that three genes may be engaged in the same pathway leading to cancer development; the schematic diagram of mutual regulation among genes is plotted in Appendix A.

GeneMINIA was used to predict the functional association network of CDK1, FAM111B and ZWINT, showing 20 potential target genes (Figure 4E). The protein-protein interaction (PPI) network map generated by STRING and similar results also showed 10 potential target proteins (Figure 4F).

### 3.10. Targeted Herbs and Drug Sensitivity Analysis of CDK1, FAM111B, and ZWINT

Because of the heterogeneity of tumors, targeted drugs showed great prospects for cancer treatment [52,53]. The online tool HERB is a high-throughput experiment- and reference-guided database of traditional Chinese medicine that was used to identify 7 potential target herbs for CDK1, 12 potential target herbs for FAM111B and 9 potential target herbs for ZWINT (Figure 5A–C). Moreover, the GSCALite database was analyzed to evaluate the correlation between medicine sensitivity and CDK1, FAM111B and ZWINT according to the Cancer Therapeutics Response Portal (CTRP) and GDSC. As shown in Figure 5D,E, CDK1, FAM111B, and ZWINT are all resistant to trametinib. In addition, the high expression levels of these genes are sensitive to 8, 2, and 11 medicines, respectively. These targeted medicines may play a role in the treatment of LUAD.

### 3.11. Verification of Prognosis Model In Vitro in Cells

The expression of the LINC00460/LINC00525-hsa-mir-338-FAM111B/ZWINT axis was verified in vitro in cell lines (Figure 6). Based on RT-qPCR results, the expression levels of LINC00460, LINC00525, FAM111B and ZWINT in three LUAD cell lines were significantly upregulated compared with BEAS-2B cells, whereas hsa-mir-338 was downregulated. The results of RT-qPCR were consistent with bioinformatics analysis.

## 4. Discussion

Despite significant advances in screening, surgery, and radiation therapy techniques, the mortality rate of LUAD is high. Elucidating the molecular mechanisms of occurrence is essential to the early warning and treatment of LUAD. It was reported that ceRNAs are involved in the occurrence and development of ovarian, gastric and liver cancer [20,21,22,23,54]. To ascertain the new tumor biomarkers expressed in LUAD, we constructed a CDK1-related ceRNA network. As a cell cycle regulatory protein, CDK1 controls cell polarity and morphology and regulates DNA replication, chromosome separation and genomic stability. However, abnormal CDK1 activity is a basis for cancer cell proliferation [55,56].

In this study, samples were divided into CDK1^low^ and CDK1^high^ groups from LUAD patients in the TCGA database, and the CDK1-related ceRNA network was obtained by calculation, screening and mutual matching. The DEmRNAs were enriched in “histone deacetylase binding”, “synaptic membrane” and “sodium ion transmembrane transporter activity”. Hence, based on survival analysis and correlation analysis, we determined the prognosis model of LUAD, namely LINC00460/LINC00525 (up-regulated)-hsa-mir-338 (down-regulated)-FAM111B/ZWINT (up-regulated). RNAs of normal lung epithelial cell line Beas-2B and three LUAD cell lines, A549, PC-9 and H1299, were determined by RT-qPCR, and we obtained consistent results.

LINC00460 and LINC00525 were upregulated in LUAD, which improved the levels of FAM111B and ZWINT via hsa-mir-338 sponge, thus promoting the occurrence of cancer. According to the clinical data from GEPIA and TCGA, LINC00460, FAM111B and ZWINT were related to a poor prognosis for patients (Appendix A). In addition, through the analysis of the clinical characteristics of patients, smoking patients were found to be associated with high expression of ZWINT, whereas the expression of FAM111B and ZWINT was independent of patient gender and age. The results of the analysis of our constructed LUAD prognostic model were consistent with the existing literature. According to the relevant reports, LINC00460 absorbed miR-149-5p, which promoted resistance to EGFR-TKI, and interleukin was up-regulated and led to epithelial-mesenchymal transition [57,58]. This was the reason why patients with EGFR mutation were difficult to treat. LINC00525 competed with the RNA-binding motif single-stranded interacting protein 2 and reduced the transcription and stability of the p21 gene by cooperating with the enhancer of the zeste 2 polycomb response complex 2 subunit [59]. Serine protease FAM111B participated in the p53 pathway by up-regulating BAG3, BCL2 and CCNB, thereby promoting cancer cell proliferation, migration and invasion [60]. Meanwhile, FAM111B regulated cyclin D1-CDK4-dependent cell cycle progression by downregulating p16 [61]. The interacting kinetochore protein ZWINT upregulated the kinetochore-microtubule attachment and spindle assembly checkpoint during meiosis. In addition, ZWINT was also a key gene leading to chromosomal instability and played an important role in homologous chromosome segregation [62,63].

In other cancers, the transcription factor Yin Yang 1 increased FAM111B transcription activity and promoted the malignant development of breast cancer [64]. In pancreatic cancer, knocking down ZWINT down-regulated NF-κB regulatory genes and effectively reduced the invasion and migration ability of cancer cells. At the same time, the deletion of ZWINT made the cells stay in G2/M phase, and the chromosome segregation process could not be performed, which increased the apoptosis of cancer cells [63].

In our study, FAM111B and ZWINT were significantly up-regulated in LUAD. Is this abnormal up-regulation regulated by some upstream mechanisms? Methylation analysis showed that the methylation levels of FAM111B and ZWINT were lower in LUAD than in normal cells. In addition, the proteins of FAM111B and ZWINT were activated by phosphorylation in LUAD. Gene mutation has been shown to be associated with the occurrence of various diseases, but in this study, it was found that mutations in FAM111B and ZWINT were not directly related to common mutations in LUAD, nor were they directly related to the OS of patients.

EMT was the main culprit in tumor spread. During the EMT process, epithelial cells lost their epithelial characteristic and cell-cell adhesion This process could be triggered by various signals. Following this process, cells acquired the ability to move and invade and metastasize to distant sites via the blood and lymphatic circulation [65]. To find out whether the expression of FAM111B and ZWINT was related to the EMT process and led to the spread of LUAD, we linked gene expression with epithelial and mesenchymal markers to monitor the progress of EMT. The results showed that FAM111B and ZWINT were negatively correlated with epithelial markers and positively correlated with mesenchymal markers. It was suggested that FAM111B and ZWINT may promote tumor proliferation through the EMT process. In addition, Li et al. reported that knock-down of FAM111B upregulated EMT-related protein E-cadherin and downregulated N-cadherin and Vimentin to promote the EMT process and breast cancer metastasis [64]. EGFR tyrosine kinase inhibitor (EGFR-TKI) is the most common targeted drug in LUAD treatment at present, but it also has drug resistance problems. There is a certain connection between the EMT process and EGFR-TKI therapy. Both intrinsic and acquired resistance to EGFR-TKIS were related to the state of EMT [66]. Byers predicted the sensitivity to erlotinib in NSCLC patients with different EMT status. Mesenchymal cells were more resistant to EGFR-inhibiting drugs, and erlotinib had greater clinical benefit in patients with EGFR wild-type tumors with an epithelial phenotype. The drug targeting of EMT also offered hope for acquired EGFR-TKI resistance [67].

FAM111B, ZWINT and CDK1 have been shown to play important roles in the development of many cancers. Ying et al. also demonstrated that ZWINT could up-regulate CDK1, but the relationship among FAM111B, ZWINT and CDK1 was not explained [68]. According to GEPIA, the top 100 genes related to FAM111B, ZWINT and CDK1 were obtained. An interesting finding was that the number one gene associated with CDK1 was ZWINT. The enrichment results of GO and KEGG also indicated a close relationship between them. Among the top five pathways of BP, CC, MF and KEGG, there were 17 duplicated pathways between CDK1 and ZWINT, which were mainly concentrated in mitosis-related regions such as chromosomes, centromeres and microtubules. The common pathways of FAM111B, ZWINT and CDK1 were also involved in mitosis. An independent pathway of FAM111B focused on DNA replication. Furthermore, the PPI network indicated the close relationship between CDK1 and ZWINT. In contrast, FAM111B was only related to spindle checkpoint protein BUB1, which interacted with CDK1, ZWINT and BUB1B.

Due to the large side effects of traditional chemotherapy, it was of great significance to find efficient, precise and low-toxic targeted anti-tumor drugs, which also met the requirements of precision medicine. With the HERB online tool and GSCALite database, we investigated the targeted drugs and drug sensitivity of CDK1, FAM111B and ZWINT, which provided a theoretical basis for further development of anti-LUAD drugs targeting CDK1, FAM111B and ZWINT.

In summary, we constructed a crucial LINC00460/LINC00525-hsa-mir-338-FAM111B/ZWINT axis, which was associated with the occurrence and development of LUAD cancer and may provide a promising therapy target for LUAD.

## Figures and Tables

**Figure 1 cells-11-01220-f001:**
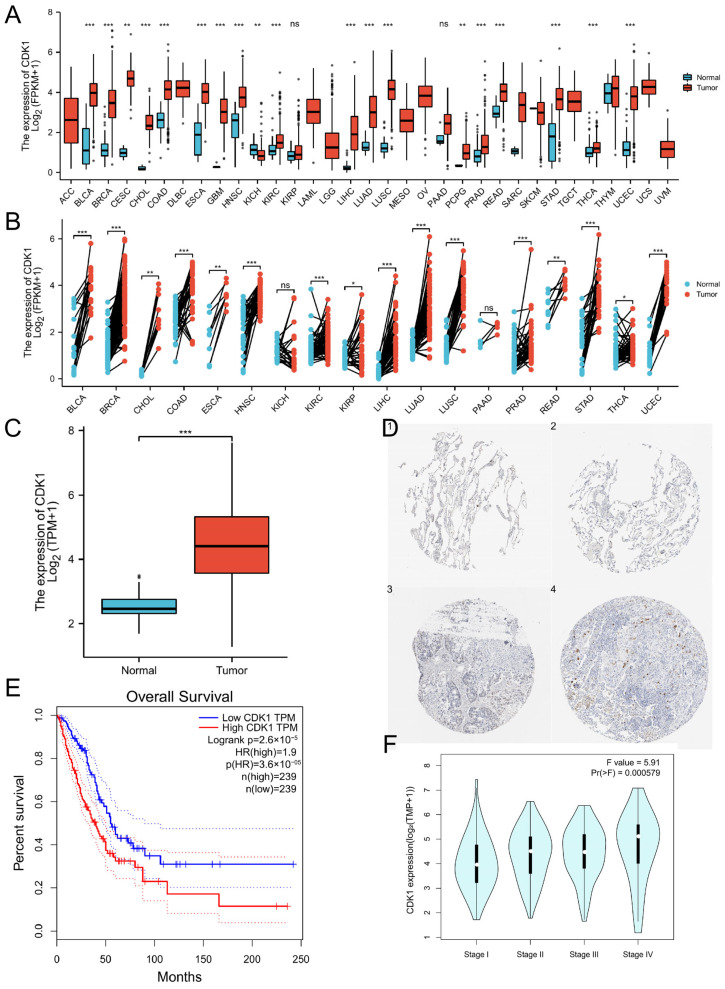
Effect of CDK1 expression on LUAD. Significance markers: ns, *p* ≥ 0.05; * *p* < 0.05; ** *p* < 0.01; *** *p* < 0.001. (**A**,**B**) Expression and pairwise analysis of CDK1 in normal tissues and pan-cancer samples. (**C**) CDK1 expression in normal tissues and LUAD. (**D**) CDK1 expression at protein level in normal tissues and LUAD (1,2 for normal tissues, 3,4 for LUAD tissues). (**E**) Correlation between CDK1 expression and patient OS. (**F**) CDK1 expression and clinical stage correlation.

**Figure 2 cells-11-01220-f002:**
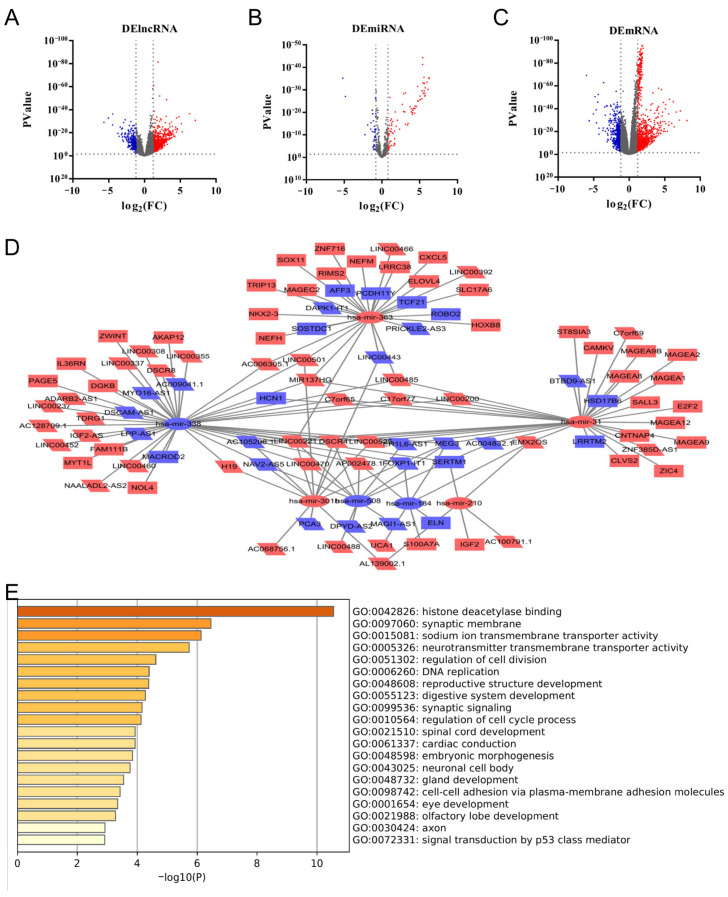
Triple regulatory network of lncRNA-miRNA-mRNA screened according to CDK1^low^ and CDK1^high^. Red means up-regulation and blue means down-regulation. (**A**–**C**) The volcano plots describe (**A**) DElncRNAs (|log_2_FC| > 1.2, *p* < 0.05), (**B**) DEmiRNAs (|log_2_FC| > 0.8, *p* < 0.05) and (**C**) DEmRNAs (|log_2_FC| > 1.2, *p* < 0.05). (**D**) CDK1-related ceRNA network. (**E**) GO enrichment analysis of DEmRNAs, bar graph of enriched terms across input gene lists, colored by *p* values.

**Figure 3 cells-11-01220-f003:**
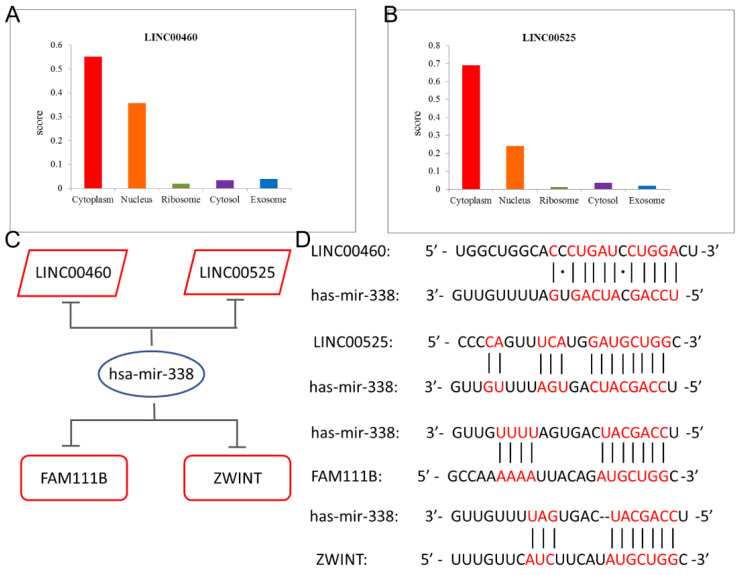
LUAD prognostic model verification of gene expression by RT-qPCR. (**A**,**B**) Subcellular localization of (**A**) LINC00460 and (**B**) LINC00525. (**C**) MiRcode, lncRNABase and TargetScan predicted the base pairing of target sites between has-mir-338 with LINC00460, LINC00525, FAM111B and ZWINT. (**D**) LUAD prognostic model.

**Figure 4 cells-11-01220-f004:**
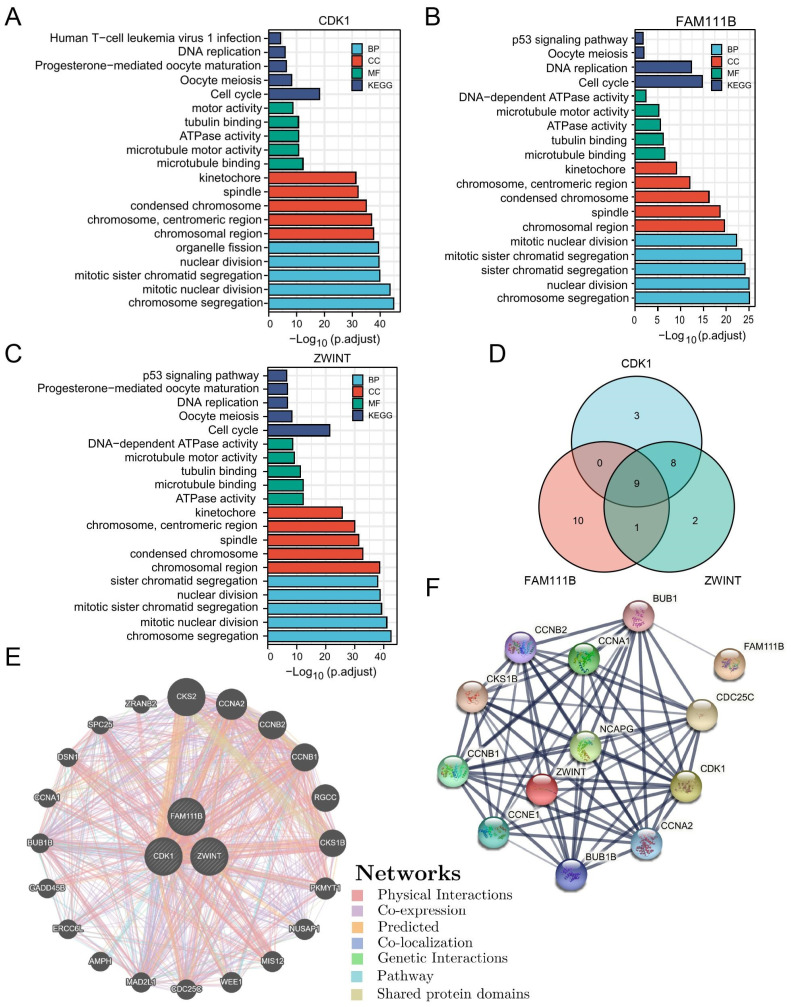
GO and KEGG enrichment analysis and interaction networks. (**A**–**C**) GO and KEGG enrichment analysis of (**A**) CDK1, (**B**) FAM111B, (**C**) ZWINT. (**D**) Venn diagram of CDK1, FAM111B and ZWINT related pathways. (**E**) Functional association networks of CDK1, FAM111B and ZWINT. (**F**) Top 10 protein interaction networks of CDK1, FAM111B and ZWINT; line thickness indicates the strength of data support.

**Figure 5 cells-11-01220-f005:**
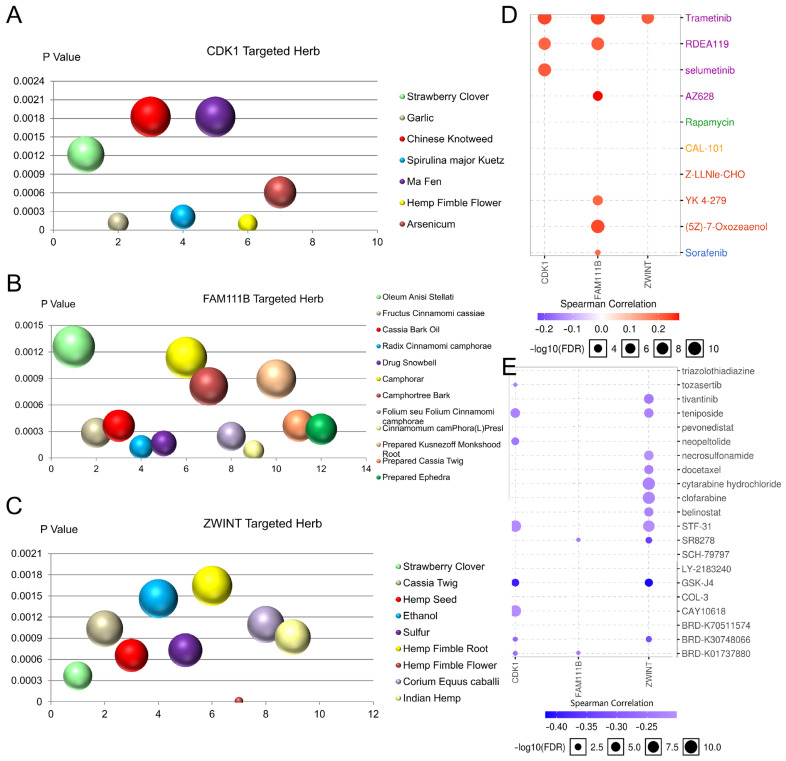
Targeted herbs and drug sensitivity analysis of CDK1, FAM111B and ZWINT. (**A**–**C**) Herbs targeting (**A**) CDK1, (**B**) FAM111B and (**C**) ZWINT. (**D**,**E**) Analysis of the correlation between CDK1, FAM111B, ZWINT and drug sensitivity based on (**D**) CTRP database and (**E**) GDSC database. Positive correlation implies that high expression of a gene is resistant to the drug, whereas negative correlation indicates sensitivity to this drug.

**Figure 6 cells-11-01220-f006:**
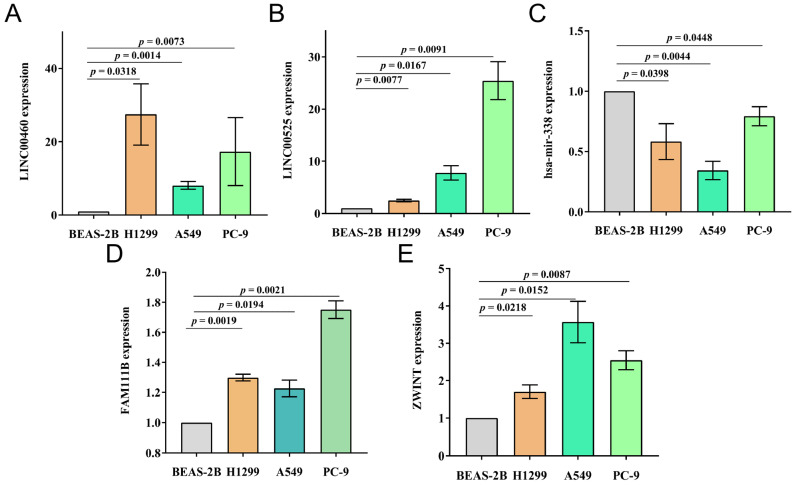
In vitro verification of prognosis model in cell lines by RT-qPCR. (**A**–**E**) The expression of (**A**) LINC00460, (**B**) LINC00525, (**C**) has-mir-338, (**D**) FAM111B and (**E**) ZWINT in normal lung epithelial cell lines (BEAS-2B) and three LUAD cell lines (H1299, A549 and PC-9).

## Data Availability

The data involved in this study were included in this manuscript and Appendix A.

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
