# Peer review of "Comprehensive Analysis of CDK1-Associated ceRNA Network Revealing the Key Pathways LINC00460/LINC00525-Hsa-Mir-338-FAM111/ZWINT as Prognostic Biomarkers in Lung Adenocarcinoma Combined with Experiments"

_cells, 2022, doi:10.3390/cells11071220_

Round 1
Reviewer 1 Report
Authors present data on CDK1-related ceRNA network analysis from TCGA suggesting potential diagnostic biomarkers or therapeutic targets for lung adenocarcinoma treatment from pathways related to LINC00460/LINC00525–hsa-mir-338–FAM111/ZWINT.
Methods and results are clear.
Language needs minor check and spelling check.
Some specific comments for section are listed below:
--Introduction needs to be improved, by adding more info on actual standard of care for lung cancer.
--Figure 1 shows descriptive data on CDK1 in TCGA. Is it possibile to summarize these data for better clarity in a summary table?
Some new analysis may help in improve the quality of data:
- Is it possible to add some data on correlation with EMT features?
- Data on immune infiltration are very interesting and promising. Can authors correlate these data with PD-L1 or immune scores available from literature for NSCLC? (see PMID: 32068166).
- Mutations of FAM111B and ZWINT are presented. Is it possible to analyze the co-mutations of these genes with other important genes related to LUAD carcinogenesis (e.g. TP53, EGFR, KRAS)?
-- Discussion can also be improved by introducing some data on epithelial to emsenchymal transition as mechanism of resistance to theraphy in LUAD.
Reviewer 2 Report
The paper offers an interesting approach resulting in the construction of an axis that should be able to detect novel therapeutic targets.
Major issue: There is not enough clinical understanding of LUAD. LUAD is presented and discussed as a homogenous cancer. In reality prognosis of the disease is varying extremely depending on driver mutation or immunogenicity scored by PD-L1-IHC. The minimum approach to publish in a high-ranked journal would be to deliver full data on mutational status, PD-L1-expression, TMB plus clinical characteristics like smoking status.
minor: the role of EMT in the resistance of EGFR-mut tumours really play a minor role. Consider involving a clinician specialized in the treatment of lung cancer
Round 2
Reviewer 1 Report
Authors improved manuscript.
I would just suggest to remove oxalipatin from drugs cited in intro since it is not used worldwide for NSCLC.
Reviewer 2 Report
Thank you very much for addressing my comments. Sorry to see that only a few driver mutations are included.
Comment 2 was maybe misunderstood as it is not addressed in the answer. It was about the role of EMT in acquired EGFR-TKI-resistance
